

# Ferulic acid attenuates high glucose-induced apoptosis in retinal pigment epithelium cells and protects retina in db/db mice

Dejun Zhu[1,*], Wenqing Zou[1,*], Xiangmei Cao[2], Weigang Xu[1], Zhaogang Lu[3], Yan Zhu[1], Xiaowen Hu[1], Jin Hu[1] and Qing Zhu[1]

[1] Department of Ophthalmology, Ning Xia Eye Hospital, People's Hospital of Ningxia Hui Autonomous Region (First Affiliated Hospital of Northwest University for Nationalities, Ningxia Clinical Research Center on Diseases of Blindness in Eye), Yinchuan, Ningxia, China
[2] Department of Pathology, School of Basic Medicine, Ningxia Medical University, Yinchuan, Ningxia, China
[3] Department of Pharmacy, People's Hospital of Ningxia Hui Autonomous Region, Yinchuan, Ningxia, China
* These authors contributed equally to this work.

Corresponding author
Wenqing Zou,
zouwenqing0706@163.com

## ABSTRACT

**Background:** Herein, we aimed to present evidence that Ferulic acid (FA), a phenolic acid, can alleviate high glucose (HG)-induced retinal pigment epithelium (RPE) cell apoptosis and protect retina in db/db mice.

**Methods:** ARPE-19 cells (a human RPE cell line) were divided into four groups: control group; HG group (30 mmol/L glucose); HG+FA group (30 mmol/L glucose and 10 mmol/L FA). Cell viability and apoptosis were detected using CCK-8 and Annexin-5 staining, respectively. Apoptosis-related markers including P53, BAX and Bcl2 were examined by RT-qPCR, western blot and immunohistochemistry. Totally, 30 male db/db mice were randomly divided into db/db group (5 ml/kg saline) and FA group (0.05 g/kg FA). After treatment for 2 months, retinal samples were subjected to hematoxylin and eosin (H&E) and Masson staining. Moreover, immunofluorescence was used to detect apoptosis-related markers. Blood samples were collected for measuring cholesterol, triglyceride (TG), low-density lipoprotein (LDL) and high-density lipoprotein (HDL) levels.

**Results:** FA treatment markedly increased cell viability and suppressed cell apoptosis of ARPE-19 cells compared to the HG-exposed group. Furthermore, FA ameliorated the abnormal expression levels of P53, BAX and Bcl2 in HG-induced ARPE-19 cells. In animal models, FA attenuated pathological changes in the retina tissues of diabetic mice. Consistent with *in vitro* models, FA significantly ameliorated the expression of apoptosis-related markers in retina tissues. Biochemical test results showed that FA reduced hyperlipidemia in diabetic mice.

**Conclusion:** Our findings suggest that FA alleviates HG-induced apoptosis in RPE cells and protects retina in db/db mice, which can be associated with P53 and BAX inactivation and Bcl2 activation.

# INTRODUCTION

Diabetic retinopathy (DR) is the most common microvascular complication in patients with type 1 or type 2 diabetes, which is roughly divided into non-proliferative retinopathy and proliferative retinopathy (*Wong et al., 2016*). It is estimated that one-third of patients with diabetes will eventually develop DR. Despite extensive research into the etiology and pathology of DR, the clinical outcomes of DR remain unsatisfactory. Thus, it is of importance to explore new drugs for DR.

High glucose (HG) has been widely recognized as a risk factor for vascular disease or permanent vision loss (*Feldman-Billard, Larger & Massin, 2018*; *Hammes, 2018*; *Lu et al., 2018*). HG can cause a variety of retinal microvascular changes, such as loss of vascular cells and increased extravasation (*Thomas et al., 2019*; *Wang et al., 2016a*). RPE, a unique epithelial cell, is an integral component of the outer blood–retina barrier, which is involved in the maintenance of normal vision (*Wang et al., 2017*). Increasing evidences suggest that HG may disrupt the normal functions of RPE cells (*Bucolo et al., 2019*; *Farnoodian et al., 2016*). Among the various cellular components of the human eye, RPE cells have been proven to be most vulnerable to HG damage. Tumor suppressor P53 regulates various cellular processes, including apoptosis. During apoptosis, P53 expression increases, translocates to the nucleus, and binds to P53 response elements in the promoter region of its target gene. Bcl-2-associated × protein (BAX) is a target gene of P53 and is involved in inducing the apoptosis processes (*Karbasforooshan & Karimi, 2018*). Studies have confirmed that P53 and BAX are closely related to RPE cell apoptosis (*Medearis et al., 2011*; *Sawada et al., 2014*). For example, P53 expression is activated in early DR in streptozotocin-induced diabetic rats (*Kovacs et al., 2011*).

FA (4-hydroxy-3-methoxycinnamic acid) that was first extracted from *Ferula foetida* are widely distributed in a variety of plants (*Ma et al., 2020*). Increasing evidences suggest that FA may improve various serious diseases such as cancers (*Grasso et al., 2020*), diabetes (*Ghosh et al., 2018*), and Alzheimer's disease (*Sgarbossa, Giacomazza & di Carlo, 2015*). For example, FA exerts a preventive effect on diabetes-related vascular damage (*Sompong, Cheng & Adisakwattana, 2015*). It is involved in regulating many biological processes including apoptosis (*Roy et al., 2014*), proliferation (*Wang et al., 2016b*), and oxidative stress (*Ghosh et al., 2018*). Recently, FA has been found to attenuate sodium iodate-induced damage in ARPE-19 cells and mouse retina (*Kohno et al., 2020*). He-Ying-Qing-Re Formula contains FA and has been found to ameliorate the phenotype in diabetic models and reduce apoptosis in retinal ganglion cells (*Zhang et al., 2018*). However, the role of FA in DR remains to be clarified. The experimental model of hyperglycemia is widely applied in the study of complications of diabetes. Genetically hyperglycemic db/db mice were used as the T2DM model (*Kobayashi et al., 2000*; *Peng et al., 2018*). In this study, we hypothesized that FA could alleviate HG-induced ARPE-19 cell apoptosis and protect retina in db/db mice.

**Table 1 Primer information for RT-qPCR.**

| Gene | Primer sequences |
|------|------------------|
| p53 | 5′-GAGGTTGGCTCTGACTGTACC-3′ (forward)<br>5′-TCCGTCCCAGTAGATTACCAC-3′ (reverse) |
| Bcl-2 | 5′-GATTGTGGCCTTCTTTGAGTTC-3′ (forward)<br>5′-ACTGAGCAGAGTCTTCAGAGACA-3′ (reverse) |
| Bax | 5′-AGCTCTGAGCAGATCATGAAGAC-3′ (forward)<br>5′-AGTTGAAGTTGCCCTCAGAAAAC-3′ (reverse) |
| GAPDH | 5′-CAAGGTCATCCATGACAACTTTG-3′ (forward)<br>5′-GTCCACCACCCTGTTGCTGTAG-3′ (reverse) |

# MATERIALS AND METHODS

## Cell culture and treatment

ARPE-19 cells (ATCC, Manassas, VA, USA) were cultured in Dulbecco's Modified Eagle Medium (DMEM; Gibco Life Technologies, Karlsruhe, Germany) plus 10% fetal bovine serum (FBS), 1% penicillin, and 1% streptomycin at the saturated humidity atmosphere of 37 °C and 5% $CO_2$ (*Dunn et al., 1996*). The culture medium was renewed every 2 days. Cells in the logarithmic growth phase were used for the experiments.

For HG treatment, when growing to approximately 80% confluence, ARPE-19 cells were then cultured in DMEM without FBS. After 24 h, the cells were incubated with 30 mmol/L HG at 37 °C for 48 h.

A total of 0.1% dimethyl sulfoxide (DMSO) was used as a co-solvent to improve the solubility and bioavailability of FA (10 mmol/L; Sigma, St. Louis, MO, USA). Moreover, a negative control group was set up to evaluate the effect of DMSO. ARPE-19 cells were treated with FA for 30 min before HG treatment. Finally, ARPE-19 cells were divided into the following groups: DMEM; DMEM+30 mmol/L HG; DMEM+30 mmol/L HG+10 mmol/L FA.

## Real-time quantitative polymerase chain reaction (RT-qPCR)

Total RNA was isolated from ARPE-19 cells or retina tissues using a TRIzol reagent kit (Invitrogen; Thermo Fisher Scientific, Inc., Waltham, MA, USA), which was reverse transcribed into cDNA by a cDNA synthesis kit (TaqMan). RT-qPCR was presented *via* the miScript SYBR Green PCR kit (Applied Biosystems, Waltham, MA, USA). The PCR conditions were as follows: one cycle for 2 min at 50 °C (initial denaturation); one cycle for 10 min at 95 °C (denaturation); 40 cycles for 15 s at 95 °C (annealing and elongation) and 40 cycles for 1 min at 60 °C (final extension). The primer sequences of target genes are listed in Table 1. GAPDH was used as a housekeeping gene. Fold changes of mRNAs were calculated with $2^{-\Delta\Delta Cq}$ method.

## Western blot analysis

Total protein was extracted from cells or retina tissues utilizing RIPA buffer (Solarbio, Beijing, China). RIPA buffer (Solarbio, Beijing, China) was used to lyse cells or retina tissues. The protein concentration was determined using the BCA protein assay kit (Pierce,

Appleton, WI, USA). Then, protein was separated with sodium dodecyl sulfate polyacrylamide gel electrophoresis (SDS-PAGE) gel, followed by transference onto polyvinylidene difluoride (PVDF; EMD Millipore, Burlington, MA, USA) membranes. The membranes were blocked with 5% skim milk for 1 h. Then, the membranes were incubated with primary antibodies including anti-p53 (1:500, ab131442; Abcam, Waltham, MA, USA), anti-Bcl-2 (1:1,000, ab182858; Abcam, Waltham, MA, USA), anti-Bax (1:1,000, ab32503; Abcam, Waltham, MA, USA), anti-Cleaved-Caspase-3 (1:500, 33199M; BSM, Shanghai, China) and anti-β-actin (1:2,000, 20536-1-AP; Proteintech, Wuhan, China) overnight at 4 °C, followed by secondary antibodies (1:6,000, ZB-5301; ZSGB-BIO, Beijing, China) for 30 min at room temperature. The proteins were visualized using an ECL kit (KeyGen Biotech Co., Ltd., Jiangsu, China). β-actin was used as an internal control. The expression of protein was measured using ImageJ software (National Institutes of Health, Bethesda, MD, USA).

## Cell viability assay

Cell viability assay was carried out using a Cell Counting Kit-8 (CCK-8; Dojindo Molecular Technologies, Inc., Tokyo, Japan). Briefly, ARPE-19 cells were seeded in 96-well plates ($5 \times 10^3$ cells/well). After treatment with HG and/or FA, 10 µL of CCK-8 solution was added to the medium and cultured at 37 °C for 1 h. The absorbance of each well at 450 nm was examined by a Microplate Reader (Bio-Rad, Hercules, CA, USA).

## Apoptosis assay

Apoptotic cells were detected using the Annexin V-FITC Apoptosis Detection Kit (BestBio, Shanghai, China). Briefly, ARPE-19 cells were seeded in six-well plates ($1 \times 10^5$ cells/well). After treatments, adherent and floating cells were harvested. The cells were resuspended with binding buffer ($1 \times 10^5$ cells/100 µL) and stained with Annexin V-FITC and propidium iodide (PI) in the dark at room temperature. After adding 500 µL of PBS, the cells were detected by flow cytometry (BD Biosciences, Franklin Lakes, NJ, USA).

## ARPE-19 immunocytochemistry staining

A total of $2 \times 10^4$/ml ARPE-19 cells grew on glass coverslips. After 5 days, the specimens were washed three times with PBS for 2 min each. A total of 4% paraformaldehyde was used to fix the cells for 15 min, followed by incubation with 0.5% Triton X-100 for 20 min and 3% $H_2O_2$ for 15 min. After blocking by goat serum (Solarbio, Beijing, China) incubation for 20 min, the cells were incubated with primary antibodies including anti-p53 (1:100, ab131442; Abcam, Waltham, MA, USA), anti-Bcl-2 (1:100, ab182858; Abcam, Waltham, MA, USA), anti-Bax (1:100, ab32503; Abcam, Waltham, MA, USA) at 4 °C overnight. PBS solution was used as a negative control. Next, cells were incubated with secondary antibody working solution at 37 °C for 30 min. DAB staining solution (Solarbio, Beijing, China) was implemented in the dark for about 10–15 min. After washing with distilled water for 5 min, the cells were counterstained with hematoxylin for 10 min. Finally, the gums were mounted. Using Image-Pro®Plus (IPP) software, 10 non-repetitive

and non-overlapping visual fields were randomly selected for each section. The average cumulative optical density (integrated optical density, IOD) value was selected and the average and standard deviation were calculated for statistical analysis.

## Animals

A total of 30 male C57BL/KsJ db/db mice (age, 8 weeks; weight, 32–36 g) and 15 age-matched C57BL/6J control non-diabetic db/m + mice (age, 8 weeks; weight, 16–18 g) were purchased from Changzhou Cavans Experimental Animal Co., Ltd. (SCXK2001-0003). All mice were housed at 23 ± 2 °C, 50 ± 10% humidity, and a 12-h light: 12-h dark cycle in accordance with the recommendations in the Guide for the Care and Use of Laboratory Animals of the National Institutes of Health. All db/db mice were randomly divided into two groups (15 mice in each group), which were daily injected intraperitoneally with physiological saline (5 ml/kg, db/db group) and FA (0.05 g/kg, FA group; Sigma, St. Louis, MO, USA) dissolved by DMSO. Male C57BL/6J control non-diabetic db/m + mice were daily injected intraperitoneally with normal saline as a normal control group. After treatment for 2 months, the body weight of all mice was measured and all mice were euthanized by intraperitoneal injection of an overdose of pentobarbital sodium (200 mg/kg). Blood samples were collected and stored at room temperature for 2 h. After centrifugation at 2,000 g for 20 min, the samples were stored at −20 °C. Eyes of mice were immediately enucleated, which were then fixed in 4% paraformaldehyde. After routinely processing, dissected retinas sample were embedded in paraffin blocks. The study was approved by the Ethics Committee of Ning Xia Eye Hospital, People's Hospital of Ningxia Hui Autonomous Region (2019085).

## Retinal immunohistochemistry staining

Retinal tissues were embedded in paraffin and then cut to approximately 5 μm thickness. Sections were then stained with Masson trichrome staining and H&E staining. Slides containing intact tissues were placed in 40 °C warm water and then dried in a 37 °C incubator. Next, the slide glass was dewaxed at a time by placing xylene-xylene-100% ethanol-100% ethanol-95% ethanol-90% ethanol-80% ethanol-70% ethanol. After rinsing, the slides were immersed in 3% $H_2O_2$ for 10 min. Then, the citric acid buffer was added to the glass slide and cooked until boiling. The sections were blocked through 5% BSA and then incubated at 37 °C for half an hour. After that, the sections were incubated with primary antibodies including anti-p53 (1:100, ab131442; Abcam, Waltham, MA, USA), anti-Bcl-2 (1:100, ab182858; Abcam, Waltham, MA, USA), and anti-Bax (1:100, ab32503; Abcam, Waltham, MA, USA) at 4 °C overnight, followed by secondary antibody at 37 °C for half an hour. After adding DAB staining solution, the sections were soaked in hematoxylin for half a minute. The counterstained sections were dehydrated, and then blocked with neutral gum. The immunohistochemical images were digitally processed by Image-Pro®Plus (IPP) software. The amount of target protein was determined in line with the color and distribution area of the dye. Five non-repetitive and non-overlapping fields of each tissue section were continuously measured under an optical microscope.

The average cumulative optical density (integrated optical density, IOD) value was selected and the average and standard deviation were calculated for statistical analysis.

## Immunocytochemistry

In the culture plate, the slides on which the cells had been crawled were immersed three times in PBS for 3 min each time, and then the slides were fixed in 4% paraformaldehyde for 15 min. After that, slides were permeated with 0.5% Triton X-100 for 20 min at room temperature. After dipping the slides in PBS, normal goat serum was added dropwise to the slides and blocked at room temperature for 30 min. After blotting off the blocking solution, the slides were incubated with primary antibodies including anti-p53 (1:100, ab131442; Abcam, Waltham, MA, USA), anti-Bcl-2 (1:100, ab182858; Abcam, Waltham, MA, USA), anti-Bax (1:100, ab32503; Abcam, Waltham, MA, USA) and anti-Cleaved-Caspase-3 (1:100, 33199M, BSM, Shanghai, China) at 4 °C overnight. Following immersing the slides in PBS, the slides were incubated with fluorescent secondary antibody at 37 °C for 1 h in the dark. The slides were incubated with DAPI in the dark for 5 min in order to label the nuclei of the cells. After the excess DAPI was washed out with PBS, the slides were mounted with a mounting solution containing an anti-fluorescent quencher, and then the images were observed under a fluorescence microscope (Leica, Wetzlar, Germany).

## Biochemical test

Serum samples were diluted with 1×PBS (1:1) and then detected cholesterol (BIV-587-100; AmyJet Scientific, Wuhan, China), triglyceride (TG; SB-2100-430; Stanbio, Boerne, TX, USA), low-density lipoprotein (LDL; A113-1-1; Nanjing Jiancheng Bio, Nanjing, China), high-density lipoprotein (HDL; SB-0599-020; Stanbio, Boerne, TX, USA) and blood glucose (BC2490; Solarbio, Beijing, China) detection kits using enzymatic double antibody sandwich methods.

## Statistical analysis

All statistical analysis was performed using GraphPad Prism software 8.0 (GraphPad Software, San Diego, CA, USA). All data are presented as the mean ± standard deviation from at least three experiments. Comparison between two groups was performed using student's $t$ test, while one-way or two-way ANOVA, followed by Dunnett's test or by Tukey's was used for multiple comparison test. $p < 0.05$ was set as the threshold of statistical significance.

## RESULTS

## FA alleviates the apoptosis of retinal pigment epithelium cells by exposure to HG

It has been widely confirmed that HG could induce RPE cell apoptosis (*Bucolo et al., 2019*; *Farnoodian et al., 2016*). Consistent with previous studies, when ARPE-19 cells were exposed to HG for 48 h, apoptotic rate of ARPE-19 cells was significantly elevated (Figs. 1A and 1B; $p < 0.0001$). Intriguingly, after treatment with FA, apoptosis of ARPE-19 cells

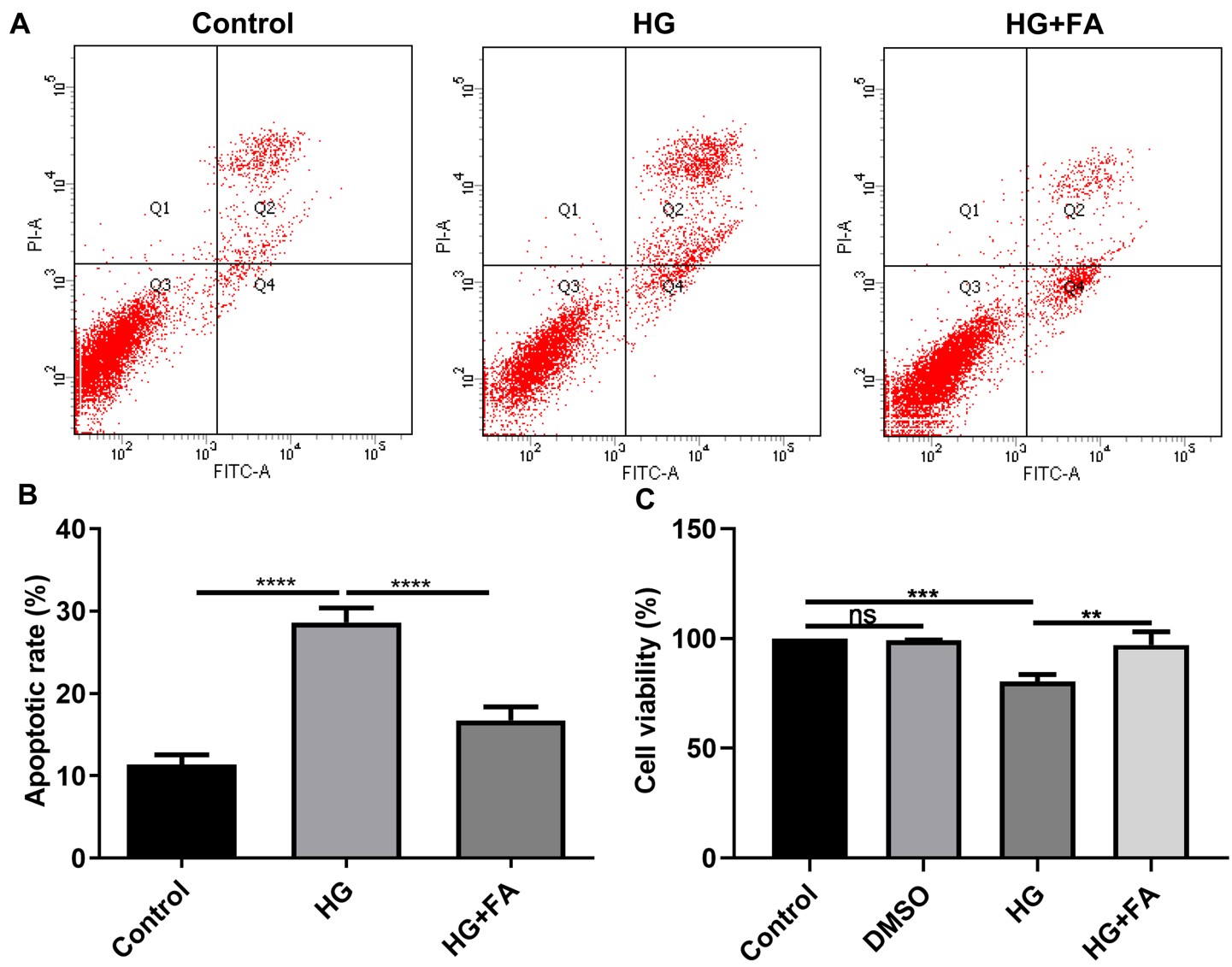

**Figure 1 FA alleviates the apoptosis of retinal pigment epithelium cells by exposure to HG.** (A) Representative of flow cytometry assay results. (B) The apoptotic rate was quantified in ARPE-19 cells treated with HG and/or FA from the average of all three experiments. (C) CCK-8 assay was used to detect the cell viability of ARPE-19 cells by exposure to HG and/or FA. **$p < 0.01$; ***$p < 0.001$; ****$p < 0.0001$; ns: no statistical significance. Three independent experiments and three replicates were used. FA, Ferulic acid; HG, high glucose.

exposed to HG was significantly inhibited (Figs. 1A and 1B; $p < 0.0001$). Furthermore, we also detected ARPE-19 cell viability using CCK-8. As shown in Fig. 1C, no effects of DMSO on cell viability were detected. But FA ameliorated the decline in ARPE-19 cell survival induced by HG ($p < 0.01$). The above findings indicate that FA could alleviate the apoptosis of ARPE-19 cells caused by the exposure to HG.

## FA reverses the expression of apoptosis-related proteins in HG-mediated retinal pigment epithelium cells

As shown in the flow cytometry assay results, FA significantly alleviated the apoptosis of RPE cells by exposure to HG. Then, we observed the expression levels of apoptosis-related

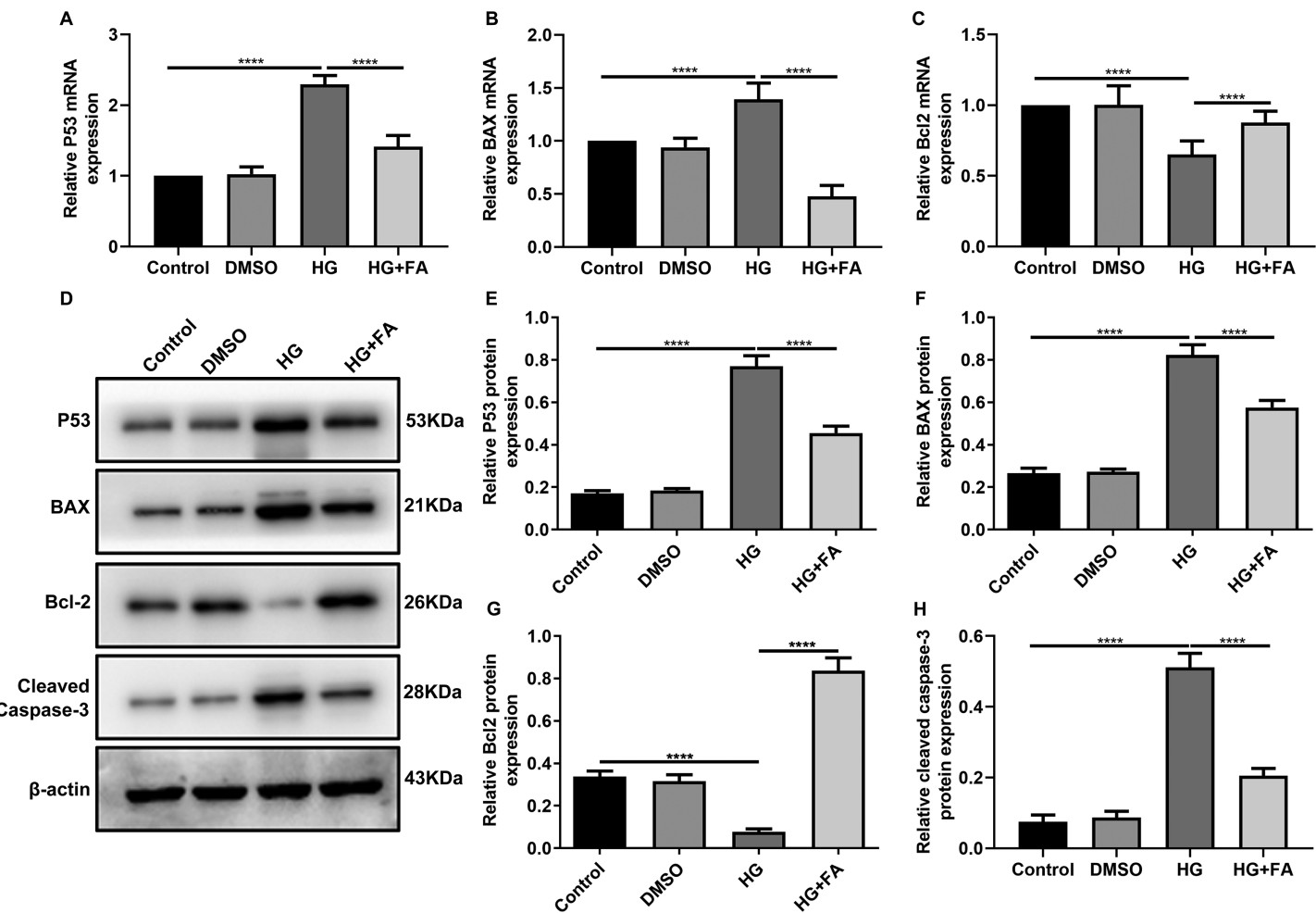

**Figure 2** **FA reverses the expression of apoptosis-related proteins in HG-mediated retinal pigment epithelium cells.** RT-qPCR results showing the mRNA expression levels of P53 (A), BAX (B) and Bcl2 (C) in ARPE-19 cells by exposure to HG and/or FA. (D) Representative images of western blot results. The protein expression levels of P53 (E), BAX (F), Bcl2 (G) and cleaved caspase-3 (H) in ARPE-19 cells by exposure to HG and/or FA. ****$p < 0.0001$. Three independent experiments and three replicates were used. FA, Ferulic acid; HG, high glucose.

markers in ARPE-19 cells exposed to HG. As shown in qRT-PCR results, P53 (Fig. 2A; $p < 0.0001$) and BAX (Fig. 2B; $p < 0.01$) were significantly activated and Bcl2 (Fig. 2C; $p < 0.01$) was markedly inactivated after HG exposure in ARPE-19 cells. We further investigated whether FA could affect the expression levels of these apoptosis-related markers. We found that, in ARPE-19 cells, FA treatment markedly ameliorated the increased expression levels of P53 (Fig. 2A; $p < 0.0001$) and BAX (Fig. 2B; $p < 0.0001$) induced by HG while ameliorated the decreased Bcl2 expression induced by HG (Fig. 2C; $p < 0.05$). Similar results were observed, as shown in western blot results. FA had a reverse effect on the expression of apoptosis-related proteins in HG-mediated ARPE-19 cells (Figs. 2D–2G). Furthermore, we found that cleaved caspase-3 expression in ARPE-19 cells using western blot. The results showed that cleaved caspase-3 expression was significantly elevated in HG-mediated ARPE-19 cells compared to controls (Fig. 2H; $p < 0.01$).

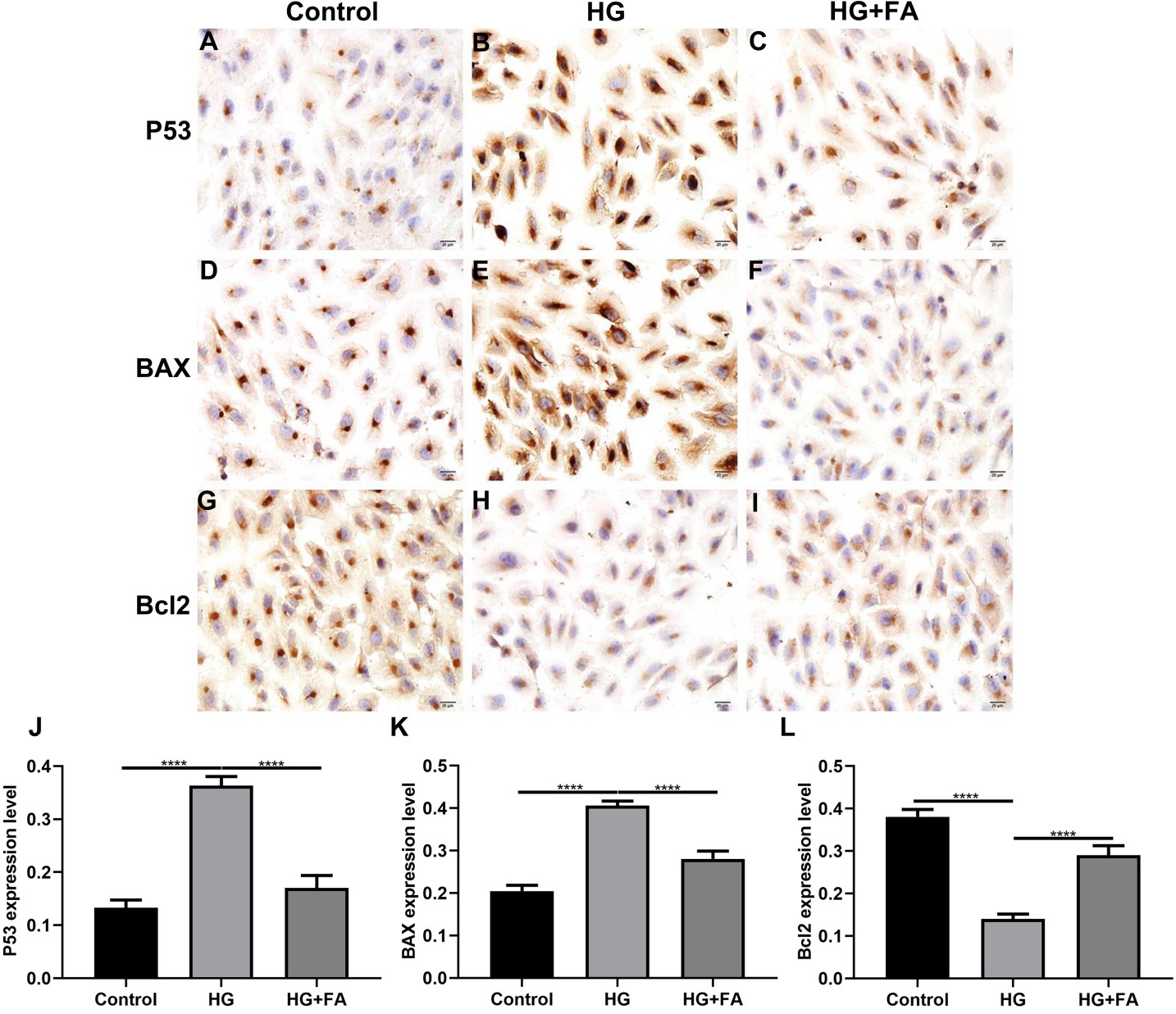

**Figure 3 Immunohistochemistry results showing the expression levels of apoptosis-related markers in ARPE-19 cells treated with HG and/or FA.** (A–I) Representative images. Expression of P53 (J), BAX (K) and Bcl2 (L) was assessed based on integrated optical density value. ****$p < 0.0001$. Three independent experiments and three replicates were used. FA, Ferulic acid; HG, high glucose. Scale bar = 20 μm.

Following treatment with FA, its expression was distinctly decreased in HG-mediated ARPE-19 cells (Fig. 2H; $p < 0.0001$). Immunohistochemistry was also performed (Figs. 3A–3I). After quantification, the expression levels of P53 (Fig. 3J; $p < 0.0001$) and BAX (Fig. 3K; $p < 0.0001$) were significantly elevated and Bcl2 expression level (Fig. 3L; $p < 0.0001$) was markedly decreased in ARPE-19 cells exposed to HG. Nevertheless, FA ameliorated the expression of apoptosis-related proteins induced by HG in ARPE-19 cells (Figs. 3J–3L).

## FA attenuates pathological changes in the retina of diabetic mice

Male db/db mice were daily injected with saline (db/db group) or FA, while age-matched wild type mice injected with saline were used as controls. After treatment for 2 months, retinal tissues of all mice (16-week db/db and 14-week db/m+ mice) were harvested and then prepared retinal sections followed by Masson staining and H&E staining. Masson staining results showed no significant increase in collagen fibers in the db/db and FA groups, suggesting that the mice were in the non-proliferative retinopathy stage (Figs. 4A–4C). The retinal thickness and structural changes were observed under a light microscopy. Compared to control wild type mice, the thickness of each layer of the retinal tissue of the model group of mice was slightly thinned, and the thickness of the photoreceptor cell layer was significantly decreased (Figs. 4D–4F). Cell edema appeared, and there was the edema of retinal ganglion cell layer and outer plexiform layer. Cells in the outer nuclear layer structure were disordered and the number of cells was reduced. However, treating db/db mice with FA restored the outer nuclear layer structure to normal levels, suggesting that FA could ameliorate cell edema of retina tissues of db/db mice. The thickness of outer nuclear layer (ONL) was measured at 2.0, 1.5, 1.0, 0.5 and 0 $\mu$m superior and inferior to the optic nerve head according to H&E staining of retinal sections. Compared with controls, the thickness of ONL was significantly increased in the db/db group (Fig. 4G). After treatment with FA, the thickness of ONL was distinctly decreased for the db/db mice. The number of positive nuclei in the ganglion cell layer (GCL) was also measured. In Fig. 4H, the number of nuclei in the GCL was markedly decreased in the db/db group than controls. However, FA treatment significantly increased the number of positive nuclei in the GCL of the db/db mice.

## FA reverses the expression of apoptosis-related proteins in retina of diabetic mice

Apoptosis is a hallmark of early DR in human and animal models. In this study, RT-qPCR results showed that, compared to control wild type mice, the mRNA expression levels of P53 (Fig. 5A; $p < 0.0001$) and BAX (Fig. 5B; $p < 0.0001$) were significantly increased and the mRNA expression levels of Bcl2 (Fig. 5C; $p < 0.0001$) were markedly decreased in retina tissues of db/db mice. Nevertheless, FA significantly ameliorated the dysregulated expression of P53, BAX and Bcl2 in retina tissues of db/db mice (Figs. 5A–5C). As shown in immunohistochemistry assay, higher P53 (Fig. 5D; $p < 0.0001$) and BAX (Fig. 5E; $p < 0.0001$) expression, lower Bcl2 (Fig. 5F; $p < 0.0001$) expression and higher cleaved caspase-3 expression (Fig. 5G; $p < 0.0001$) were found in retina tissues of db/db mice compared to control wild type mice. FA treatment ameliorated the expression of these apoptosis-related proteins in retina tissues of diabetic mice (Figs. 5D–5S). Moreover, we performed immunofluorescence staining of these proteins in retina tissues of diabetic mice. Figures 6A–6I showed the representative images. Following quantification, the results showed that the dysregulated expression of these apoptosis-related proteins including P53 (Fig. 6J), BAX (Fig. 6K), Bcl2 (Fig. 6L) and cleaved caspase3 (Fig. 6M) in retina tissues of diabetic mice was markedly ameliorated after FA treatment. Our western blot results also confirmed that FA treatment significantly decreased the expression of P53,

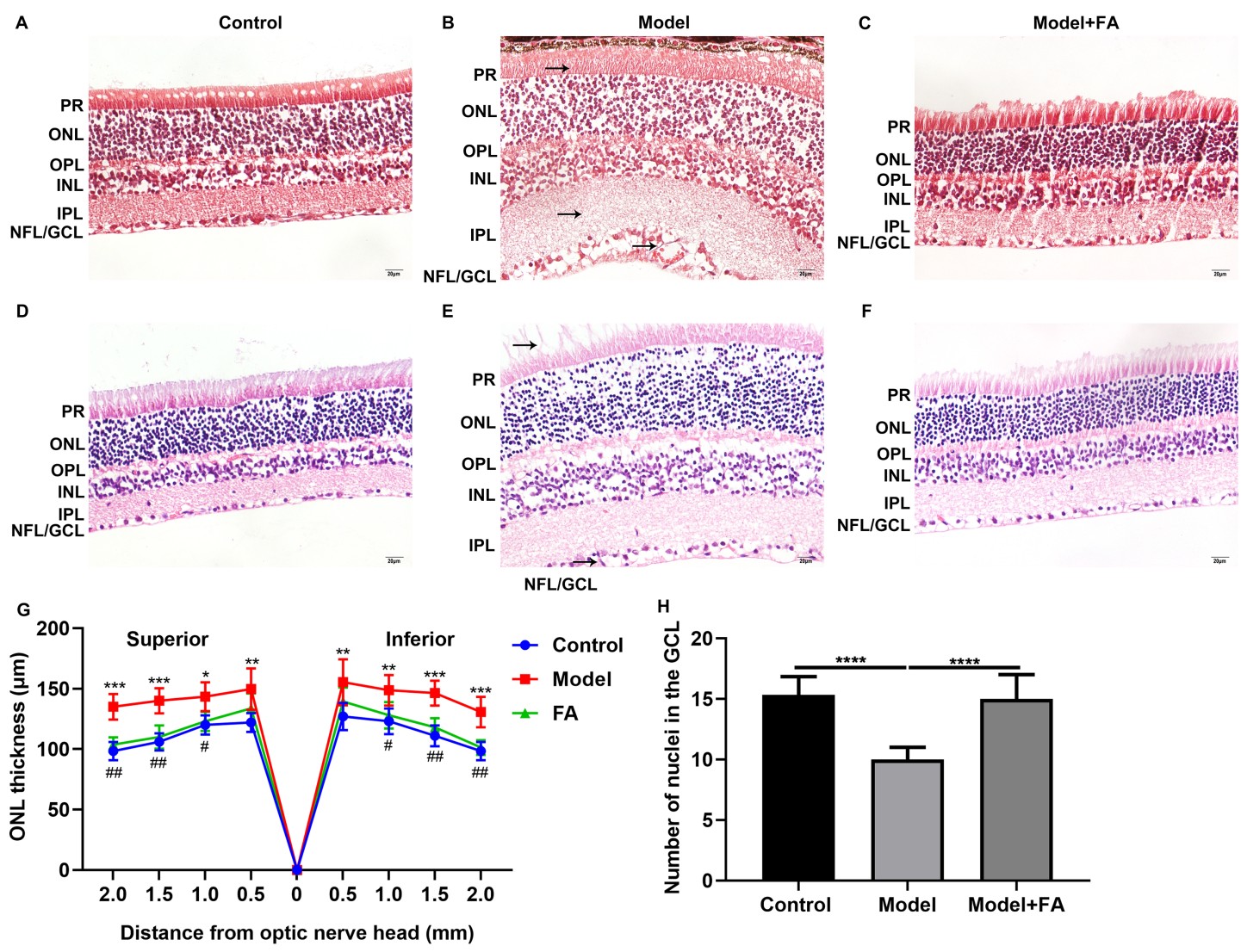

**Figure 4 FA ameliorates retinal abnormalities in db/db mice.** (A–C) Representative images of Masson staining. (D–F) Representative images of H&E staining. (G) The thickness of outer nuclear layer (ONL) was calculated at 2.0, 1.5, 1.0, 0.5 and 0 μm superior and inferior to the optic nerve head. The statistics were carried out through two-way ANOVA followed by Tukey's multiple comparisons test. The statistical comparisons were presented between the control *vs* model group and the model *vs* FA group. (H) The number of nuclei in the ganglion cell layer (GCL) was also measured. There were three groups: 14-week wild type mice injected with saline (a control group); 16-week db/db mice daily injected with saline (a db/db group); 16-week db/db mice daily injected with FA (a FA group). Different retinal layers (photoreceptor layer (PR); outer nuclear layer (ONL); outer plexiform layer (OPL); inner nuclear layer (INL); inner plexiform layer (IPL); nerve fiber layer/ganglion cell layer (NFL/GCL)) were labeled and the pathological changes were pointed out by arrows. Compared to control group, $*p < 0.05$, $**p < 0.01$, $***p < 0.001$, $****p < 0.0001$; compared to model group, $\#p < 0.05$, $\#\#p < 0.01$. Three mice were used and three sections of each mouse were counted. FA, Ferulic acid. Scale bar = 20 μm.

BAX, and cleaved caspase3 as well as increased the expression of Bcl2 in retina tissues of diabetic mice (Figs. 6N–6R).

## FA inhibits hyperlipidemia of diabetic mice

Serum cholesterol, TG, LDL, blood glucose and HDL levels in db/db mice were determined 2 months after treatment. Compared to control non-diabetic mice, serum cholesterol

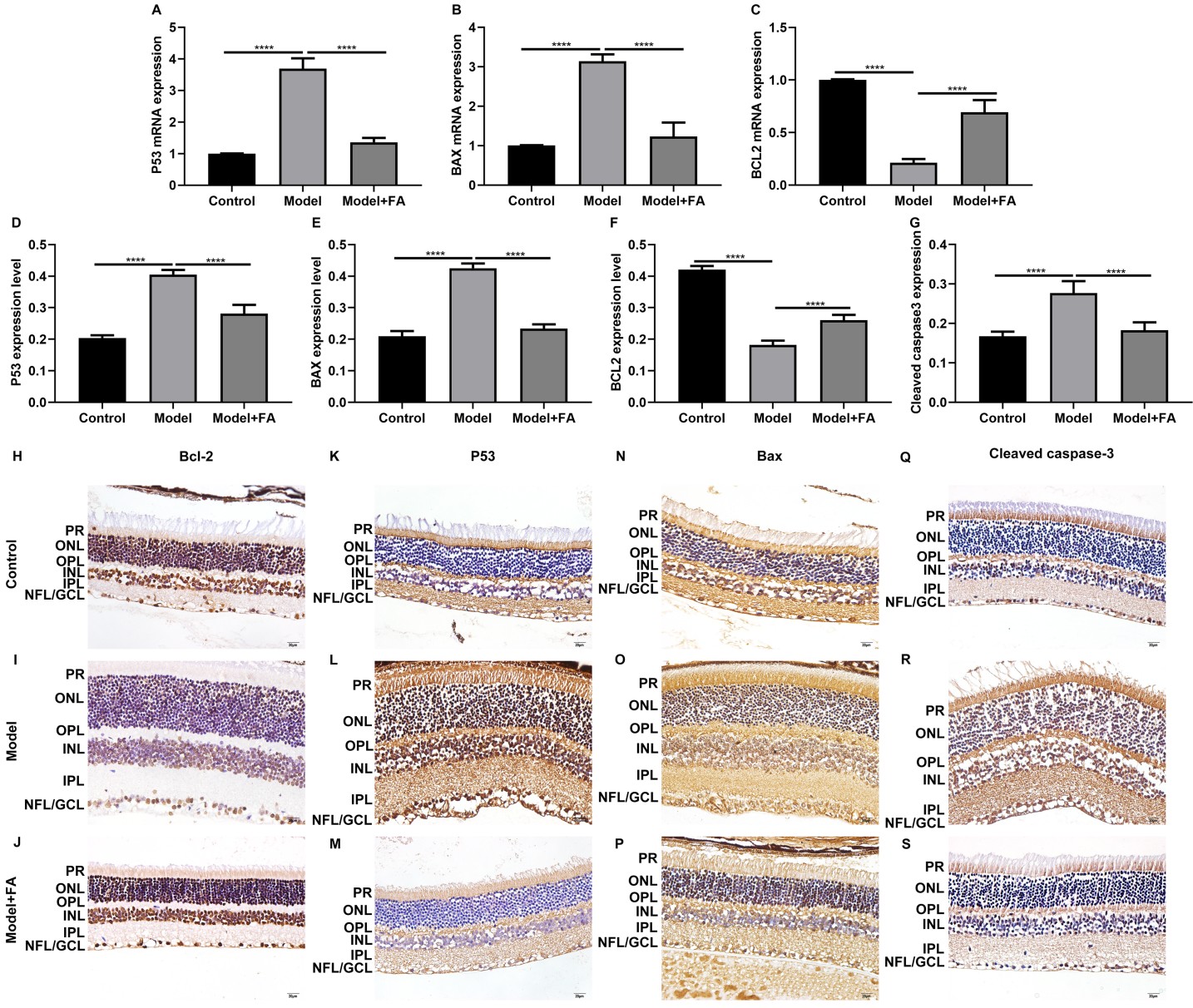

**Figure 5** **FA reverses the expression of apoptosis-related proteins in retina of diabetic mice.** RT-qPCR results showing the mRNA expression levels of P53 (A), BAX (B) and Bcl2 (C) in retina of diabetic mice treated with FA or not. Immunohistochemistry results showing the expression levels of P53 (D), BAX (E), Bcl2 (F) and cleaved caspase-3 (G) in retina of diabetic mice treated with FA or not based on integrated optical density value. (H–S) Representative images of immunohistochemistry. ****$p < 0.0001$. Three mice were used and three sections of each mouse were counted. FA, Ferulic acid. Scale bar = 20 µm.

(Fig. 7A; $p < 0.01$), TG (Fig. 7B; $p < 0.01$), LDL (Fig. 7C; $p < 0.01$) and blood glucose levels (Fig. 7D; $p < 0.0001$) in diabetic mice were significantly elevated; furthermore, serum HDL levels in db/db mice had a lower level than control non-diabetic mice (Fig. 7E; $p < 0.01$). FA treatment can inhibit diabetes-induced hypertriglyceridemia in db/db mice (Figs. 7A–AE). Also, we measured the body weight of three groups. The results showed that the body weight of db/db mice was significantly higher than that of controls (Fig. 7F;

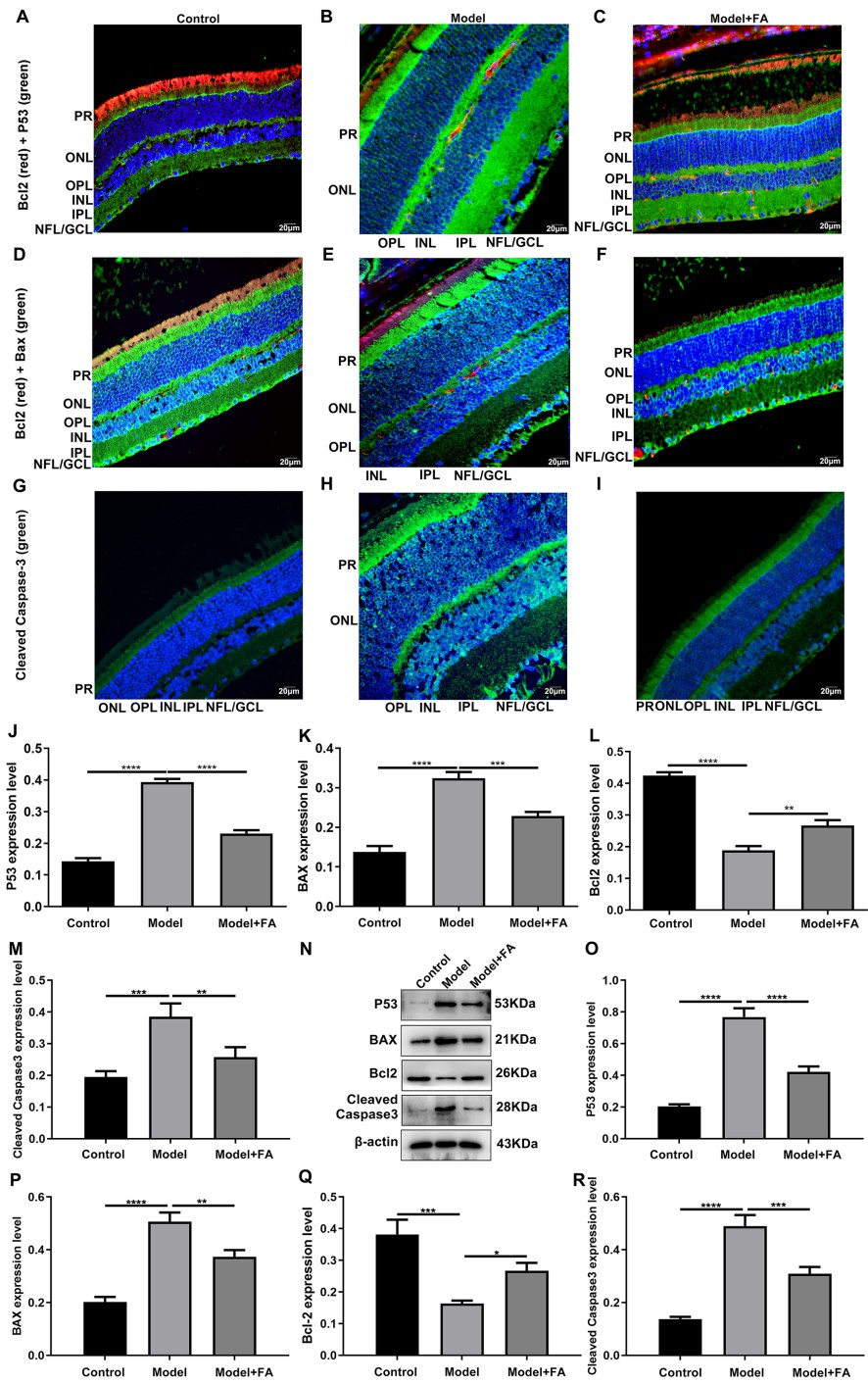

**Figure 6  Immunofluorescence staining showing the changes in expression levels of apoptosis-related proteins in retina tissues of diabetic mice treated with FA or not.** (A–I) Representative images of immunofluorescence staining. Quantitative results of P53 (J), BAX (K), Bcl2 (L) and cleaved caspase-3 (M) in retina of diabetic mice based on integrated optical density value. (N) Representative images of western blot results. The protein expression levels of P53 (O), BAX (P), Bcl2 (Q) and cleaved caspase-3 (R) in each group. $^*p < 0.05$; $^{**}p < 0.01$; $^{***}p < 0.001$; $^{****}p < 0.0001$. Three mice were used and three sections of each mouse were counted. FA, Ferulic acid. Scale bar = 20 μm.

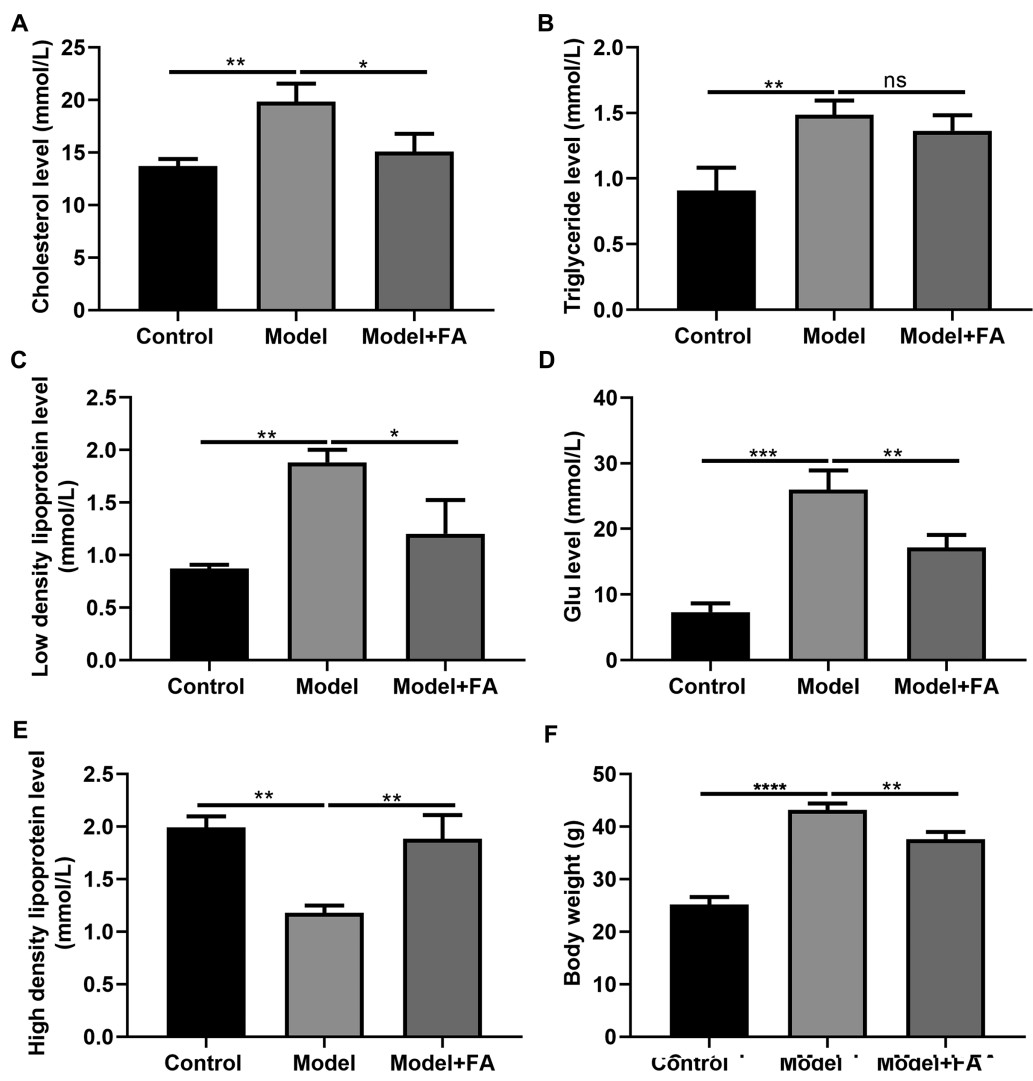

**Figure 7 FA inhibits hypertriglyceridemia in db/db mice.** (A) Cholesterol; (B) triglyceride; (C) low-density lipoprotein; (D) blood glucose; (E) high density lipoprotein; (F) body weight. $^*p < 0.05$; $^{**}p < 0.01$; $^{***}p < 0.001$; $^{****}p < 0.0001$; ns: no statistical significance. There were 15 mice in each group. FA, Ferulic acid.

$p < 0.0001$). After treatment with FA, the body weight of db/db mice was distinctly decreased ($p < 0.01$).

## DISCUSSION

DR is often observed in patients with a long history of diabetes, which can eventually lead to blindness. In this study, our evidence suggested that FA could reduce the apoptosis of ARPE-19 cells induced by HG. Administration of FA to db/db mice can ameliorate diabetic retinal edema and attenuate the development of DR. This effect may be attributed to the anti-apoptotic effect of FA. Mechanistically, FA could reverse the abnormal expression of apoptosis-related proteins, which can prevent RPE cell apoptosis and protect the retina from damage caused by diabetes.

DR is closely related to prolonged hyperglycemia, which can seriously damage the retina, leading to vision loss and blindness (*Hammes, 2018*). HG has been shown to impair endothelial function in animal and human studies. In the early stage of DR, hyperglycemia can cause retinal vascular leakage, which swells the retina and the macula. In the later stage, abnormal new blood vessels can grow on the surface of the retina, and scar tissue and tiny exudative particles can form in the retina. Apoptotic death of retinal cells including RPE cells directly affects the visual function (*Stehouwer, 2018*; *Wan et al., 2015*). In this study, when ARPE-19 cells were treated with HG (30 mmol/L) for 48 h, cell viability was greatly reduced and cell apoptosis was suppressed compared to the viability of control cells. Intriguingly, FA could ameliorate cell viability and apoptosis of ARPE-19 cells exposed to HG. *Bogdanov et al. (2014)* has found that there is a significant increase in Tunnel positive apoptotic cells in 16-week db/db mice. Furthermore, the role of lowering glucose in ameliorating the phenotype in db/db mice has been highlighted, finding lower apoptosis in the GCL of db/db mouse retina when mice are fed a restrictive diet (*Bogdanov et al., 2014*). Besides ARPE-19 cells, FA increases cell viability and reduces apoptosis of hepatocytes and cardiomyocytes exposed to HG (*Song et al., 2016*). FA improves HG-induced oxidative stress in H9C2 cells (*Salin Raj, Swapna & Raghu, 2019*). Mechanistically, we found that FA can prevent HG-induced increase in BAX (pro-apoptosis) and decrease in Bcl-2 (anti-apoptosis). In addition, FA can suppress the activation of p53 caused by HG in ARPE-19 cells. P53 is known to be a factor involved in retinal apoptosis (*Ao et al., 2019*). Consistent with previous studies, in HG-stimulated ARPE-19 cells, Bcl-2 expression is decreased, while BAX and P53 expression is increased (*Liu et al., 2017*, *2018*; *Zhang & He, 2019*). Furthermore, $H_2O_2$ treatment could also result in a significant increase in BAX expression and a decrease in Bcl-2 expression in ARPE-19 cells (*Hu et al., 2019*). Thus, P53 and BAX activation and Bcl-2 inactivation are significantly associated with RPE cell apoptosis during the development of DR. More importantly, FA could inhibit HG-induced RPE cell apoptosis by P53/BAX/Bcl-2 pathway.

In this study, db/db mice were treated with FA for 2 months. Our results showed that FA significantly ameliorated retinal edema of db/db mice. Retinal edema is a common feature of DR (*Toyoda et al., 2016*). Consistent with *in vivo* models, FA could improve the expression of apoptosis-related proteins in retina of diabetic mice according to RT-qPCR, immunohistochemistry and immunofluorescence.

Strict control of hyperlipidemia is still the main strategy for DR treatment, but quite a few patients still have difficulty meeting the recommended goals (*Donzelli et al., 2018*; *Itoh et al., 2018*; *Ueshima et al., 2016*). Current DR treatments, including laser photocoagulation and anti-VEGF drugs, can significantly reduce the incidence of severe vision loss (*Itoh et al., 2018*). However, existing therapies have not been successful in preventing vision loss, and they are accompanied by troublesome side effects and serious complications (*Olsen, 2015*). Thus, there is an urgent need for novel alternative drug therapies based on the pathophysiology of DR. Hyperlipidemia is characterized by high levels of cholesterol, TG and LDL and lack of HDL, which is associated with different stages of diabetes. Reducing LDL and VLDL levels and increasing HDL levels have been proven to be effective in primary and secondary prevention of vascular complications of

diabetes. Furthermore, in a hyperglycemic environment, hyperlipidemia increases the development of DR (*Kowluru et al., 2016*). In this study, our results suggested that FA treatment significantly inhibited serum cholesterol, TG and LDL levels and increased HDL levels in db/db mice, indicating that FA had an inhibitory effect on diabetes-induced hypertriglyceridemia.

In this study, the data revealed that FA alleviated HG-induced apoptosis in RPE cells and protected retina in db/db mice, which could be in association with P53 and BAX inactivation and Bcl2 activation. However, the specific mechanism of FA in inhibiting diabetes-induced RPE apoptosis needs to be explored in the longer term.

## CONCLUSION

In this study, we found that FA can alleviate HG-induced RPE cell apoptosis and viability. Moreover, FA could improve the dysregulated expression levels of apoptosis-related markers including P53, BAX and Bcl2 in HG-induced ARPE-19 cells and retina tissues of db/db mice, indicating that FA alleviated diabetes-induced cell apoptosis by P53 and BAX inactivation and Bcl2 activation. Also, FA reduced hyperlipidemia in diabetic mice. Thus, FA has the potential to become a drug candidate for the treatment of DR, which deserves further exploration.

## ABBREVIATIONS

| | |
|---|---|
| **RPE** | Retinal pigment epithelium |
| **DR** | Diabetic retinopathy |
| **FA** | Ferulic acid |
| **HG** | High glucose |
| **TG** | Triglyceride |
| **LDL** | Low-density lipoprotein |
| **HDL** | High-density lipoprotein |
| **RT-qPCR** | Real-time quantitative polymerase chain reaction |

### Funding

This work was funded by the Natural Science Foundation of Ningxia Hui Autonomous Region (2018AAC03176), the Ningxia Blind Eye Disease Clinical Medical Research Center Innovation Platform Project, and the Ningxia Hui Autonomous Region Key R&D Plan Project in 2021 (2021BEG03110). The funders had no role in study design, data collection and analysis, decision to publish, or preparation of the manuscript.

### Grant Disclosures

The following grant information was disclosed by the authors:
Natural Science Foundation of Ningxia Hui Autonomous Region: 2018AAC03176.
Ningxia Blind Eye Disease Clinical Medical Research Center Innovation Platform Project.
Ningxia Hui Autonomous Region Key R&D Plan Project: 2021BEG03110.

## Competing Interests

The authors declare that they have no competing interests.

## Author Contributions

- Dejun Zhu conceived and designed the experiments, performed the experiments, prepared figures and/or tables, and approved the final draft.
- Wenqing Zou conceived and designed the experiments, authored or reviewed drafts of the paper, and approved the final draft.
- Xiangmei Cao performed the experiments, analyzed the data, prepared figures and/or tables, and approved the final draft.
- Weigang Xu performed the experiments, prepared figures and/or tables, and approved the final draft.
- Zhaogang Lu performed the experiments, authored or reviewed drafts of the paper, and approved the final draft.
- Yan Zhu analyzed the data, authored or reviewed drafts of the paper, and approved the final draft.
- Xiaowen Hu analyzed the data, authored or reviewed drafts of the paper, and approved the final draft.
- Jin Hu analyzed the data, prepared figures and/or tables, and approved the final draft.
- Qing Zhu performed the experiments, prepared figures and/or tables, and approved the final draft.

## Animal Ethics

The following information was supplied relating to ethical approvals (*i.e.*, approving body and any reference numbers):

The study was approved by the Ethics Committee of Ning Xia Eye Hospital, People's Hospital of Ningxia Hui Autonomous Region (2019085).

## Data Availability

The raw measurements are available in the Supplemental Files.

## Supplemental Information

Supplemental information for this article can be found online at http://dx.doi.org/10.7717/peerj.13375#supplemental-information.

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
