# Peer review of "Ferulic acid attenuates high glucose-induced apoptosis in retinal pigment epithelium cells and protects retina in db/db mice"

_PeerJ, doi:10.7717/peerj.13375_

## Round 0.1 · original submission · Major Revisions

Please consider carefully the comments from the reviewers. Some extra experimental work is required to support the main conclusions.

·

Basic reporting

1. The English language should be improved to ensure that an international audience can clearly understand your text. It is advised that the authors seek help from a professional scientific writer to meet scientific and English language standards.
Some examples where the language could be improved include:
Line 34: “using CCK-8 and Flow Cytometry.” Apoptosis is not detected by flow cytometry but by Annexin-5 staining that is visualized using flow cytometry.
Line 39: “blood samples were collected for detection” should be replaced by blood samples were collected to measured… levels
Line 48: “FA alleviatd HG-induced cell apoptosis in RPE cells,” should be FA alleviated HG-induced apoptosis in RPE cells
Line 56: “It is estimated that one third of patients with diabetes will eventually occur DR.” should be replaced by “…will eventually develop DR.”
Line 63: “is an integral component of the blood-retina barrier” should be “is an integral component of the outer blood-retina barrier”
Line 281: “Mechanically” should be replaced by “Mechanistically”
Line 131: “cells were seeded in a dish for cell climbing” does the author want to say that cells grow on glass coverslips?
Line 163: “Alcohol” should be replaced by “ethanol”
Line 172: “Immunofluorescence” should be replaced by “immunocytochemistry”
Line 182 “in order to nucleate” should be “in order to label the nuclei of the cells”


2. Although the authors clearly demonstrated the effect of Ferulic acid on RPE and retinal cells. There are few weaknesses related to data gathering, unsupported conclusion and figure quality that need to be improved before acceptance. It is unclear how protein expression was quantified. Moreover, the retinal layer thickness should be quantified by measuring the length of the different layers at different regions of the retina and presented using a spidergram. Adding to the manuscript the quantification of tunnel or cleaved caspase-3 positive cells in retinal sections would substantially support the authors´ claims.

3. Figures
- Figure 1 on panel B “Apoptostic rate” should be “Apoptotic rate”; on panel C “viability(%)” should be “viability (%)”
- Figure 2 panels A, B and C “expersion” should be replaced by “expression”; panels E, F and G “experssion” should be “expression”. It is not clear how protein expression was measured and what is depicted on the graphics.
- Figure 3. It is not clear how protein expression was measured and what is the unit of the values.
- Figure 4 and 5. Should be combined to see side-by-side the effect of the treatments on the wildtype and db/db mice. All retinal sections should be presented with the same orientation, e.g., photoreceptors in the top part. Labels should be added to easily identify the different retinal layers, e.g., ONL, INL, GCL. Arrows or arrowheads should be used to pinpoint the alteration mentioned in the text. Scale bars should be made visible.
- Figure 6. Similar as before, it should be clear how the protein quantification was performed, and the units of the values should be added. Retinas should be presented in the same orientation in a straight position and retinal layers should be properly labelled. Scale bars should be made visible.
- Figure 7. Similar as indicated to figure 6.

4. Raw data
- Images of the full western blot membrane should be provided.

Experimental design

Materials and methods
- Line 97: The concentration of FA used on the in vitro studies should be indicated in mol/L?
- There is no information about the cDNA synthesis and the PCR reaction condition. This information should be added to the manuscript.
- The author should indicate how the protein extraction was performed and how protein expression was measured.
- Information about the DAB staining should be added.
- It is extremely important to make clear if animals were injected daily with FA for 2 months. This is not very clear on the material and methods section.
- Did the authors use dissected retinas or all eyes for the analysis? They should indicate how tissues were processed.
- Line 166: Please indicate the type of serum and the concentration used.
- Please indicate the mounting medium used.
- Line 187: which enzymatic methods were used?

Validity of the findings

Results:
- Line 230-234: To support the authors´ statement retinal thickness should be measured. Moreover, cell edema and edema of ganglion cell layer are not obvious and therefore the small alterations on the morphology should be indicated by arrows on the figure.
- Line 253–259: the author measured a range of different biochemistry parameters, however maybe the most important one glucose was not measured.

Additional comments

The manuscript by Zhu et al demonstrated that Ferulic acid protects RPE cells from high-glucose induced apoptosis. Moreover, experiments done using db/db mice, a widely used model of type 2 diabetes mellitus, suggested that Ferulic acid protects retinal cells from high-glucose induced apoptosis. The authors should better connect the two parts of the manuscript. The title just reflects the in vitro part of the work, once on the mouse studies changes in the RPE were not analysed.

Reviewer 2 ·

Basic reporting

The manuscript entitled “Ferulic acid protects retinal pigment epithelium cells from high glucose-induced apoptosis” by Zhu et al. looks to expand upon the known role of Ferulic acid (FA) in ameliorating apoptosis in response to high glucose (HG) in an ARPE-19 model and a db/db mouse model. The authors present some interesting data on a relevant topic and provide a balanced discussion. The manuscript, in some areas, is underdeveloped and the flow between the use of the ARPE-19 line and the db/db mouse model needs to be clearer. In particular, as the authors chose not to analyses the RPE from this model but only the retina.
The authors should refrain from make statements such as “as expected” and “obvious”. If a statement such as “It has been widely confirmed” is used (line 198), please include some citations to this information. A number of small changes to improve flow of text are required. As an example, line 203 “Above findings indicated” should be changed to “The above findings indicate”. Line 204 “the apoptosis of RPE cells by exposure to HG” should be changed to “the apoptosis of RPE cells caused by the exposure to HG”. Line 207 “As shown in Flow Cytometry” should be “As shown in the Flow Cytometry”. Line 223 “changes in retina” to “changes in the retina”. Line 278 “Except ARPE-19: to “Besides ARPE-19”. Line 312 “In this study, both in vivo and in vitro models, our findings revealed” to “In this study, the data from both our in vivo and in vitro models revealed that:.
Discuss in the introduction the role of RPE in diabetic retinopathy (DB) in more detail. As an RPE in vitro system is implemented, clarification of its relevance in DB and thus to the subsequent experiments carried out in the db/db mouse model are required. Also discuss further the roles of FA in attenuating induced damage models of RPE and Retina. As an example, recently FA was found to attenuate induced damage in an ARPE-19 and mouse retina (Kohno et al. 2020, Scientific reports). There are also a number of papers which discuss the role of FA in treating HG, the introduction should include more information on this already well-established role (Sompong et al. 2015, PloS ONE). Lastly, He-Ying-Qing-Re Formula contains FA and has been found to ameliorate the phenotype in a diabetic model, reducing apoptosis (Zhang et al. 2018, Journal of Ethnopharmacology). Additions do not have to be extensive and if deemed more appropriate can be added to the discussion.

Experimental design

The manuscript by Zhu et al. brings together a spontaneous diabetic mouse model of type 2 diabetes and a high glucose induce RPE cell apoptosis model. However, the authors do not adequately link these two models together.
-The title only focus on one aspect of the author’s paper (RPE) and does not tie in FA treatment in the db/db mouse model.
-Furthermore, only retinal tissue from the db/db mice are analyzed. In addition, collection and evaluation of the RPE would have helped link the data from the two models.
- Evaluation of the 16-week-old treated db/db retina needs to be done more thoroughly, compared to the controls, to assess the validity of the authors conclusion “FA attenuated pathological changes in the retina of diabetic mice”. I suggest the authors – Quantify total retinal thickness, ONL thickness, and INL thickness. Furthermore, the authors may wish to quantify the number of cells in the GCL, it should be reduced in the db/db mouse model (Bogdanov et al. 2014, PLoS One).
-The authors should strengthen their findings on the role FA pays in ameliorating apoptosis in the diabetic mouse retina by performing and quantifying Cleaved-caspase 3 staining and/or the Tunnel assay to show that in the FA treated retinal sections that apoptosis is reduced. A significant increase in Tunnel positive apoptotic cells is found in 16 week db/db mice (Bogdanov et al. 2014, PLoS One). Furthermore, the Bogdanov et al. paper highlights the role of lowering glucose in ameliorating the phenotype in db/db mice, finding lower apoptosis in the GCL of db/db mouse retina when mice were fed a restrictive diet. The authors may wish to further discuss this relevant finding in the context of their own work.
-The authors could strengthen their findings by carrying out a western blot for P53, BAX and BCL2 in the control, Model and FA mouse groups. To verify optical density quantifications.
-Please state in the results section and the figure legends the exact age at which mice were analyzed. 16 weeks db/db and 14weeks db/m+?
The authors provide no functional data e.g. ERG for A, B and C wave to show therapeutic efficacy. This would have greatly strengthened the authors findings.

Validity of the findings

Some statements made by the authors are not backed up by experimental findings. This is mainly in reference to the db/db mouse model and amelioration of its phenotype by FA.
-It is hard to clearly read the scale bar length from the figures, please state in the legends.
Figure 1. Please indicate that figure 1A is an example from one experiment, while Figure 1B is the average of all 3 experiments.
Figure 3: In the legend and results please make it clear that this is optical density quantifications. Please clearly state the method used to analyze the data. How were regions of interested created (ROIs), was background signal corrected for, were images taken at the same intensity, etc
-Figures 4, 5, 6, 7. The authors describe that the retinal thickness of each layer was slightly thinner and that the ONL thickness was significantly decreased. This general description does conform to the expected phenotype observed in the db/db mouse retina. However, the authors provide no quantifications for any of their statements. Furthermore, all their images of the db/db untreated mouse retina appear thicker than the control or FA groups, assuming images were taken at the same magnification and from similar areas of the retina. This may be due edema but then the number of nuclei in a row could be counted as an additional approach.
-Figure 5: Could the authors highlight the areas of edema in B. Please refrain from using statements such as “obvious”. Please quantity the number of cell nuclei in the ONL, do not just state that the number of nuclei are reduced without providing quantifications. “FA restored it to normal levels” What is “it”. Further in Line 292 the authors state that “FA significantly ameliorated retina edema” not quantification of this was done.
-Figure 6, 7: Could the authors clarify in text and in the legend that the quantification of protein expression levels is from analysis of immunohistochemistry (6) and fluorescence (7). E.g. Please state optical density of images was quantified. Please include a thorough description of the methodology used to analyze this data. How were regions of interested created (ROIs), was background signal corrected for, were images taken at the same intensity, etc
-Figure 8B: The model and FA group have highly over lapping error bars. I performed an unpaired t-test using the author’s raw data, and found no significant difference. Please correct.
Furthermore, could the authors double-check the statistics from other analysis.
-For clarity please include the Cat numbers of the Biochemical assays used from Stanbio.
-Please state exact information about equipment used. E.g. what type of fluorescence microscope.
-How were retina sections prepared. E.g. what percentage of PFA , were the cryoprotected with sucrose, were they stored in OCT compound.
-Consider using the wording attenuated or ameliorated instead of reversed throughout the manuscript.

·

Basic reporting

Overall Zhu and colleagues present a well-written manuscript in which authors introduce and discuss their work based on a list of recent literature references and show the beneficial effects of Ferulic acid (FA) in reverting diabetes-mediated retina effects. Concerning the figures, please consider to perform the following alterations:

1 - There are some grammar typos that must be corrected. Specifically, the Y axis of Figure 1B have "Apoptostic rate" instead of "Apoptosis rate". Also, on Y axis of Figure 2, "expersion" must be corrected to "expression" and the word "The" must be removed from all the Y axis of this Figure.

2- Figure 4 must be improved with a legend indicating the name of the different tissues that are observed on the figure to better understand the pathological alterations detected.

3 - The same for Figure 5.

4 - On Y axis of Figure 6A-C, remove the word "The".

5 - Regarding the figures with results obtained from animals, please remove the word "Model" when referring to the diabetic animals and simply use db/db or diabetes.

6 - In all figures legends authors must include the "n" in which were performed the experiments.

7 - Authors should improve the last paragraph of introduction. Presently is only focused in the in vitro study and do not mention the work that will be performed in animals.

8 - Also, authors should explain why the work was perform in this specific animal model of diabetes.

9 - Authors included a supplementary pdf file with representative western blots bands but do not explain what that means.

Experimental design

In the present manuscript, authors were solely focused on investigating the possible beneficial effects of FA on the apoptosis process occurring on a scenario of diabetic retinopathy. For that the study was conducted in both in vitro and in vivo models of diabetes. The experimental design was well thought and authors used appropriate methodologies to validate their hypothesis. Still, authors should consider the following comments:

1 - On materials and methods section, lines 97-98, authors say "Moreover, a negative control group was set up to evaluate the effect of DMSO". However, looking at figures, DMSO possible effects were only assessed when evaluating the mRNA and proteins expression levels of apoptosis-related proteins. It would be important to also evaluate DMSO effects on cells viability. Please consider to perform this additional experiment.

2 - Did authors performed any osmotic control to the glucose concentration used in the study? If not, please consider to perform this additional control on cells viability.

3 - Regarding the work with animals, authors must provide animals full characterization with body weight and glycemia values.

4 - In the abstract, authors say "...an HG+FA group (30mmol/L glucose and 2 mg/ml FA)", and in the material and methods section, line 100, we can read "...DMEM+30mmol/L HG+0.05g/kg FA." It seems that there is a mistake. Please verify this.

5- Please explain on the materials and methods section why the FA concentration used in the study was the same for the cells and for the animals and include a reference to justify it.

6 - Did authors used DMSO to dissolve the FA that was administered to the animals? Please include this information on materials and methods section.

Validity of the findings

Overall, authors validated their hypothesis by demonstrating the positive effects of FA in reverting the apoptosis process in both in vitro an in vivo models of diabetes. Authors do not propose any mechanisms.

---

## Round 0.2 · Major Revisions

There are several major points that need to be clarified, as indicated by the reviewers. Particular attention must be given to apoptosis-related parameters.

·

Basic reporting

The authors were able to improve the manuscript, although the English levels can still be considerably improved. Although the authors did perform most of the requested changes, the manuscript still presents several issues:

Line 76/77: "Thus, inhibition of apoptosis in RPE cells has become an approach in the treatment of DR". Is inhibition of apoptosis in RPE cells an approved treatment for DR? If not sentence should be correct otherwise it might mislead the reader.

Related to the figures, as requested before the authors should increase the size of the lettering in the figures mainly of the scale bars. The spidergram presented now on figure 4G should be replaced by one similar to presented in Figure 3 of the following manuscript: DOI: 10.1371/journal.pone.0154779 "Limited ATF4 Expression in Degenerating Retinas with Ongoing ER Stress Promotes Photoreceptor Survival in a Mouse Model of Autosomal Dominant Retinitis Pigmentosa." It should be clear that the thickness of the different retinal layers (ONL, INL, etc) are measured at different distances from the optic nerve.

It is not clear how many retinas/samples were used for the different analysis, eg, how many samples were on figure 5D. Accordingly to the raw data only 3. Moreover, the raw data is already pre-analysed, presenting the mean and the SD, and not the individual values.

Experimental design

The research question is clear and the material and methods were clearly improved. However, still not always clear how many samples/animals were used in each analysis.

Validity of the findings

no comment

Additional comments

The authors were able to improve the quality of the present manuscript however a few points still need to be clarified before publication.

Reviewer 2 ·

Basic reporting

The authors have, in part, addressed some of the points made in the first round of revision. In particular, the authors have improved the use of citations to back up some of the previous statements made in the first version of the manuscript, in addition to improving their introduction.
However, the authors still have not sufficiently addressed several points from the first round of review, including the following points:
• Significantly the authors have not clearly shown apoptosis in retinal section and its amelioration upon FA treatment. They state they show this in Figure 2D in their rebuttal, but this addresses Cleaved –Caspase staining in ARPE-19 “Thanks. We have performed and quantified Cleaved-caspase 3 staining to show that in the FA treated retinal sections that apoptosis is reduced. The results have shown in Figure 2D.” The comments from the initial review report still stand “The authors should strengthen their findings on the role FA pays in ameliorating apoptosis in the diabetic mouse retina by performing and quantifying Cleaved-caspase 3 staining and/or the Tunnel assay to show that in the FA treated retinal sections that apoptosis is reduced. A significant increase in Tunnel positive apoptotic cells is found in 16 week db/db mice (Bogdanov et al. 2014, PLoS One).”

• The authors state that they have “performed immunohistochemistry and immunofluorescence to examine and quantify P53, BAX and BCL2 proteins in the control, Model and FA mouse groups. We think that the results are reliable.” While the staining may be reliable, one way to significantly strengthen the data is to show this change in levels through western blotting for P53, BAX and BCL2.

• The addition of Figure 4G is a step in the correct direction for showing the required thickness measurements, which would adequately clarify the differences between the Control, db/db and FA groups. However, these would be best shown as individual layer spider grams (as an example see Alves et al. 2019, IJMS Figure 5G-I). In its current format, Figure 4G is hard to read and also does not show the “significant” differences between the Control, db/db and FA groups the authors discuss between lines 260-274. The authors do not provide any statistics for these points. The comments from the initial review report still stand “Evaluation of the 16-week-old treated db/db retina needs to be done more thoroughly, compared to the controls, to assess the validity of the author's conclusion “FA attenuated pathological changes in the retina of diabetic mice”. I suggest the authors – Quantify total retinal thickness, ONL thickness, and INL thickness. Furthermore, the authors may wish to quantify the number of cells in the GCL, it should be reduced in the db/db mouse model (Bogdanov et al. 2014, PLoS One).”

• Line 235: The above findings indicate that FA could alleviate the apoptosis of RPE cells by exposure to HG. Change to: The above findings indicate that FA could alleviate the apoptosis of RPE cells “caused” by the exposure to HG. – This was also commented on in the first round of review. RPE cells should be changed to ARPE-19 cells.

• Figure 3J, please correct the overlap of the significant bars.

• Discussion Line 336-341. Here the authors talk about the treatment of the db/db mice and the last sentence states “Therefore, we have reason to state that FA could suppress diabetic RPE cell apoptosis”. However, the authors do not analyze the RPE in the db/db mice and therefore can not make this statement. In turn, Line 357-359 also can not be said. As previously stated if you wish to make such statements, the authors should also analyze the RPE.

Experimental design

na

Validity of the findings

na

·

Basic reporting

No comment.

Experimental design

Authors revised the manuscript accordingly to the suggested comments but, it remains unclear the number of animals that were used to perform each experiment. The sentence "Each experiment was repeated three times" is not very clear and if it means that were only used three animals of each group, this is quite insufficient when it refers to animal studies. Please check this.

Validity of the findings

No comment.

Additional comments

Overall, authors followed the reviewers comments and suggestions and present an improved version of the manuscript.

---

## Round 0.3 · Major Revisions

Two reviewers have the opinion that the changes made to the paper are minor and did not improve its scientific quality. I urge you to reconsider their comments and perform the necessary alterations.

·

Basic reporting

The authors should seek a professional scientific writer to help then with the editing of the manuscript.

The raw data presented are not truly raw but mean values. And the spidergram is not correctly presented, there is no mention in which area the measurements were performed. As indicated before the measurements should be performed in different areas/distances of the optic nerve of the retina.

The author did not provide a satisfactory reply to the previous comments.
The manuscript presents the (same) weakness as the lastest version.

Experimental design

Figure 6. No BCL2 is visible on B much likely because the retinal section misses the outer segments, the image should be replaced.

Clearly staining for apoptotic markers could be improved. They present a high background. Using fluorescence labelling coupled with confocal imaging will help the author to better present their data, and at the same time count the number of apoptotic cells (e.g. tunnel positive or cCasp3 positive)

Validity of the findings

It is not clear how the authors only used 3 retinas per analysis when then have a total of 15.
The raw data is not present, only the means and SD are presented.

Reviewer 2 ·

Basic reporting

The authors made some effort to revise the comments from the last round of review. However, additional issues still remain:
• I thank the authours for trying to carry out the Cleaved-caspase staining (Figure XQ-S). However, I am not fully convinced about the quality of this staining and would suggest the authours to carry out immunoflurecens. I would expect to see minimal Cleaved caspase staining in control compared to db/db mice, please see Figure 6 of the following manuscript: https://journals.plos.org/plosone/article/figure?id=10.1371/journal.pone.0097302.g006 Please also quantify the number of positive nuclei in the GCL.

• The Authors still did not addressed the comments from the last review rounds and do not show any negative data that the tried. Previous comment “The authors state that they have “performed immunohistochemistry and immunofluorescence to examine and quantify P53, BAX and BCL2 proteins in the control, Model and FA mouse groups. We think that the results are reliable.” While the staining may be reliable, one way to significantly strengthen the data is to show this change in levels through western blotting for P53, BAX and BCL2.” The authours stated in their rebuttal to this comment “Reply: Thanks for the wonderful comment. Because the eyeballs of mice are so small that enough proteins cannot be extracted, it is difficult to detect the expression of P53, BAX and BCL2 proteins by western blot. Therefore, we performed immunohistochemistry and immunofluorescence. Our immunohistochemistry and immunofluorescence confirmed that FA could ameliorate the apoptosis of diabetic retinal tissues.” There reply is unfounded it is possible to westernblot for all of the indicated proteins in mouse retina. As an example see: https://www.jneurosci.org/content/33/5/2205#sec-2


• Could the Authors state how the statistics were carried out for Figure 4G, Multiple T-Test? For clarity, the provide statistical comparisions mentioned in the legend are between the Control vs Model group and the Model vs FA group?

• From my previous round of review “Discussion Line 336-341. Here the authors talk about the treatment of the db/db mice and the last sentence states “Therefore, we have reason to state that FA could suppress diabetic RPE cell apoptosis”. However, the authors do not analyze the RPE in the db/db mice and therefore can not make this statement. In turn, Line 357-359 also can not be said. As previously stated if you wish to make such statements, the authors should also analyze the RPE.” Thank you for removing the sentence started “Therefore, we have reason to state that FA could suppress diabetic RPE cell apoptosis”. However, the authours did not address the issue with the subsequent lines now 356-58. You did not look at the RPE from the in vivo model and there for cannont make this statement.

Experimental design

N/A

Validity of the findings

N/A

Additional comments

N/A

·

Basic reporting

The conclusion subsection of the abstract should be written using the present tense, i.e. "Our findings suggest that FA alleviates HG-induced apoptosis in RPE cells and protects
retina in db/db mice, which can be associated with P53 and BAX inactivation and Bcl2 activation."

Experimental design

No comment.

Validity of the findings

No comment.

Additional comments

The paper was greatly improved after considering all the reviewers comments and suggestions.

---

## Round 0.4 · Minor Revisions

Below you can find the reviewer's comments. Particular attention is to be paid to the raised statistical concerns.

·

Basic reporting

The manuscript and the writing were marginally improved. Although the authors replied to all my previous comments, they still do not provide robust and trusty data.

Experimental design

Authors should make very clear what is the "n" in each experiment. How many independent experiments and how many replicates were used. For example, the spidergram, how many mice were used and how many sections were counted.
It is quite a surprise that the authors have (only) 3 values for all the experiments/groups.

Please rotate the retinal pictures to have the same orientation and label the different layers.

Validity of the findings

Line 277/581 and Fig 4H, What are the "Positive cells"? These cells are positive for?

Lines 305: Remove "therapeutic effect". It is better to describe and discuss the data rather than have general statements.

Line 581-589: What was the number of mice/sections used?

Figure 5G, the middle picture seems to have excessive staining of cCasp3, the entire retina seems positive.

Additional comments

'no comment

---

## Round 0.5 · accepted · Accept

In my view, all remaining issues pointed out by the reviewer were adequately addressed and the manuscript was revised accordingly. Therefore, the amended version is acceptable now.